# Influence of sample temperature and environmental humidity on measurements of benzene in ambient air by transportable GC-PID

Cristina Romero-Trigueros[1], Marta Doval Miñarro[2], Esther González Duperón[1], Enrique González Ferradás[1]

[1]Chemical Engineering Department, School of Chemistry, University of Murcia, 30071, Murcia, Spain
[2]Chemical and Environmental Engineering Department, Technical University of Cartagena, Paseo Alfonso XIII, 52, 30203 Cartagena, Murcia, Spain

*Correspondence to*: Marta Doval Miñarro (marta.doval@upct.es)

**Abstract.** Calibration of *in situ* analysers of air pollutants is usually done with dry standards. In this paper, the influence of sample temperature and environmental humidity on benzene measurements by gas chromatography coupled with a photo ionisation detector (GC-PID) are studied. Two reference gas mixtures (40 µg/m³ and 5 µg/m³ nominal concentration benzene in air) were subjected to two temperature cycles (20 ℃ / 5 ℃ / 20 ℃ and 20 ℃ / 35 ℃ / 20 ℃) and measured with two identical GC-PIDs. The change in sample temperature did not produce any significant change in readings. Regarding ambient humidity, the chromatographs were calibrated for benzene with dry gases and subjected to measure reference standards with humidity (20% and 80% at 20 ℃). When measuring a concentration of 0.5 µg/m³ benzene in air, the levels of humidity tested did not produce any significant interference in measurements taken with any of the analysers. However, when measuring a concentration of 40 µg/m³, biases in measurements of 18% and 21% for each analyser, respectively, were obtained when the relative humidity of the sample was 80% at 20 ℃. Further tests were carried out to study the nature of this interference. Results show that humidity interference depends on both the amount fractions of water vapour and benzene. If benzene concentrations in an area are close to its annual limit value (5 µg/m³), biases of 2.2% can be expected when the absolute humidity is 8.6 g/cm³ –corresponding to a relative humidity of 50% at 20 ℃-. This can be accounted for in the uncertainty budget of measurements with no need for corrections. If benzene concentrations are above the annual limit value, biases become higher. Thus, in these cases, actions should be taken to reduce the humidity interference, as an underestimation of benzene concentrations may cause a mismanagement of air quality in these situations.

## 1 Introduction

Benzene is one of the species regulated in the European Union (EU) in air quality by Directive 2008/50/EC (EU, 2008). Its harmful health effects have been studied during the last five decades (Bahadar et al., 2014; Gist and Burg, 1997; Haley, 1977; Smith, 2010). Evidence for an association with childhood leukaemia (D'Andrea and Reddy, 2016a) and alterations in

hematologic and liver profiles in adults (D'Andrea and Reddy, 2016b) is lately growing. The major source of benzene in cities is gasoline, as it is one of its components as well as a product of its combustion (von Schneidemesser et al., 2010).

Continuous measurements of benzene concentrations in air monitoring stations are carried out using automated pumped sampling with in situ gas chromatography. This analytical method must comply with the requirements of Standard EN 14662-3:2015 (EN, 2015). After separation of the organic components, they are usually quantified by a flame ionisation detector (FID) – not selective for BTEX (benzene, toluene, ethylbenzene and xylenes) - or by a photo ionisation detector (PID) – a more selective one for aromatics. Liaud et al. (2014) recently compared the performance of a transportable gas chromatograph coupled with a photo ionisation detector (GC-PID) to a thermo-desorption device coupled to GC-FID. This study revealed that the transportable GC-PID was the most sensitive technique allowing an efficient separation and quantification of the six BTEX compounds in 12 minutes.

PIDs consist of an ultraviolet lamp which produces high energy photons that collide with the molecules of the target gas and ionise them, as long as their ionisation potentials are below the energy of the photons (Peng et al., 2010). PIDs are compact yet they exhibit high performance, featuring excellent response characteristics and detection sensitivity on a ppb scale (Peng et al., 2010). As main drawbacks, they have been reported high power consumption and susceptibility to humidity (Barksy et al., 1985; Peng et al., 2010).

There are several works in the literature that assess the performance of the PID to measure volatile organic compounds (Adamia et al., 1991; Barksy et al., 1985; Coy et al., 2000; LeBouf et al., 2013; Mouradian and Flannery, 1994). However, most of them are related to measurement of species in work places. Occupational concentrations of pollutants are usually at mg/m$^3$ levels whereas regulated concentrations of ambient air pollutants are, in most cases, at μg/m$^3$ levels. Particularly, the European annual limit value for benzene is 5 μg/m$^3$ (EU, 2008). These low concentrations require a specific assessment of the influence of the environmental conditions on measurements of benzene in air. In a recent paper by the authors, the influence of pressure on benzene measurements by GC-PID was studied (Romero-Trigueros et al., 2016). In this work, we focus on the influence of sample temperature and humidity on ambient measurements of benzene obtained with a commercial transportable GC-PID. Although the instrument that we test also measures toluene, ethylbenzene and xylenes, in this work, we only focus on benzene as it is currently the only VOC with limit values in air quality.

## 2 Materials and methods

### 2.1 Experimental set-up

The influence of sample temperature and humidity was studied with two identical type-approved on-site BTEX Syntech Spectras GC955 chromatographs, named analysers I and II, equipped with photo ionisation detectors. The air sample is forced through a built-in preconcentration system. Hydrocarbons are preconcentrated on Tenax GR, thermally desorbed and

separated on an AT-5 capillary column (15 m length x 0.32 mm i.d. x 1 μm HELIFLEX coating). The two analysers are preconfigured to identify as benzene the signals detected by the PID in the windows 176–212 s and 148–182 s, respectively. Each measurement cycle lasts for 15 min.

An in-house designed dynamic dilution system devoted to test analyser performance was used for the generation of known concentrations of benzene in zero air at controlled conditions. This chamber has been described elsewhere (Romero-Trigueros et al., 2016) and only a brief description is given here.

Purified compressed ambient air was used as zero gas. Humidity was added to a portion of the zero air by means of an in-house designed humidifier (Figure 1a). The humidifier consists of a glass sphere with two lateral inlets (1 and 2) for the zero air to enter and exit the humidifier, respectively. Water is pumped through a glass tube (3) inserted in a third inlet located at the bottom of the sphere (4). The water impacts the top of the sphere and falls down creating a wet film on the walls which favours the mass transfer. The water is collected at the bottom of the sphere (5) and taken to a container provided with thermal insulation where it is stored. When the system is working, the water from the container is pumped to the humidifier through a thermostatic bath where its temperature is readjusted. The whole system is leak-tested and relative humidities up to 99% are attainable depending on the temperatures of the zero air and the humidifying water, and the ratio of zero air flowrate through the humidifier to the flowrate of dry zero air. In Figure 1b a schematic of the humidifying system integrated in one of the lines of dry zero air is shown.

Sample relative humidity and temperature were measured with a Testo 645 thermo hygrometer (precision ± 1%). From these values, the humidity mixing ratio, W -the ratio of the actual mass of water vapour present in the sample to the mass of the dry air- was derived. The water vapour flow rate, $q_{H2O}$ (l/min), added to the flow of zero air, $q_z$ (l/min), was calculated using Eq. (1), where 28.8 and 18 are the molar masses of dry air and water, respectively, in g/mol. It is important to know this flow so that the final concentration of benzene in the reference mixture is calculated accounting for it. Thus, any decrease in benzene measurements when measuring wet samples cannot be attributed to the dilution effect of water vapour.

$$q_{H2O} = W \cdot q_z \cdot \frac{28.8}{18} \qquad (1)$$

A high concentration mixture of benzene in nitrogen (1000 μg/m$^3$ nominal concentration, 5% expanded uncertainty) from a gas cylinder (Abelló Linde, Spain) was mixed with the humidified zero air to attain the experimental concentrations required. The gas mixture was certified by Abelló Linde according to Standard ISO 6141:2007. The flow of gas in each branch, Figure 1b, was controlled and measured with Bronkhorst mass flow controllers (0–0.4 l/min range for the benzene in nitrogen mixture, and 0–12 l/min for the zero air). The mass flow controllers are at least annually calibrated against a Gilian Gilibrator (a NIOSH primary standard air flow calibrator) available in our lab.

Sample temperature was changed and controlled by flowing the sample through an in-house made thermostatic bath. The whole piping system right after the thermostatic bath –which included the MFCs, the humidifier, the mixing area and the sampling manifold- was set up inside a thermally controlled chamber to maintain the sample temperature. Sample inlet pressure was set up to be equal to normal atmospheric one (101.3 ± 0.2 kPa). Control of ambient conditions is pivotal to ensure that

changes on measurements are due to the effect of the environmental parameter under test and not to other environmental conditions.

## 2.2 Experimental methods

### 2.2.1 Calibration

The analysers used in this work have three different calibration options, namely, a linear calibration using a least squares regression; a calibration line forced through the origin; and finally, a non-linear regression. All three calibration options were tested with eight different mixtures of benzene in air with concentrations ranging from 0.0 to 47.2 µg/m$^3$ (0.0, 0.65, 2.60, 5.20, 10.4, 15.6, 26.3, 36.7 and 47.2 µg/m$^3$). Thus, three calibration curves were obtained and the squared sum of residuals of the concentration tested was obtained for each calibration. The lowest sum of squares (1.16) was obtained with the non-linear

(quadratic) calibration, followed by the least squares regression (1.66) and the linear regression forced through the origin (1.78). Therefore, the quadratic option was chosen every time the analysers were calibrated. Calibration was performed at 20 ℃ and using dry gases.

### 2.2.2 Design of experiments to study the influence of sample temperature on the analyser readings

In order to study the influence of sample temperature on the analysers, two reference gas mixtures (40 µg/m$^3$ and 5 µg/m$^3$

nominal concentration) were measured with analysers I and II at different temperatures. Two temperature cycles were performed. First cycle was performed at 20 ℃, then changed to 5 ℃ and back to 20 ℃ (temperature control precision ±2 ℃). The second one was performed at 20 ℃, then changed to 35 ℃ and back to 20 ℃ again. Once the sample temperature was stabilised 4 measurements were taken at each concentration level.

### 2.2.3. Design of experiments to study the influence of humidity on the analyser readings

**2.2.3.1 First set of experiments**

As a first approach to the subject, the tests described in Standard EN 14662-3:2005 were carried out after calibrating the analysers according to Section 2.2.1. These tests were carried out before the release of the 2015 version of the Standard and this is why they were performed according to the previous version. These tests consist of measuring a reference mixture of 0.5 µg/m$^3$ nominal concentration benzene in air with a relative humidity of 20% and 80% at 20 ℃ and comparing the results.

Standard EN 14662-3:2005 defines the influence of the relative humidity by means of coefficient $b_{rh}$, calculated as:

$$b_{rh} = \frac{|\bar{C}_{rh,max} - \bar{C}_{rh,min}|}{|\bar{C}_{rh,max} + \bar{C}_{rh,min}|/2} \cdot 100 \tag{2}$$

Where $\bar{C}_{rh,max}$ and $\bar{C}_{rh,min}$ are the average of 6 consecutive readings when measuring the reference gas mixture (0.5 µg/m$^3$ nominal concentration benzene in air) with an 80% and 20% relative humidity, respectively, at 20 ºC. Standard EN 14662-3:2005 establishes that $b_{rh}$ has to be lower than 4%.

The tests were repeated with a reference mixture of 40 µg/m$^3$ nominal concentration benzene in air with the same relative
humidities and temperature. A significant difference in readings was noticed when working with the high concentration reference mixture with both analysers. Further tests with analyser I were then performed to study in depth this phenomenon. It is worth noting that the test for evaluating the influence of ambient humidity in the new version of the Standard EN 14662, from 2015, has substantially changed. The current version requires testing benzene at the annual limit value with relative humidity of 80% at 20 ºC and comparing the measurements with those obtained with the same benzene amount fraction with
no humidity. Parameter $b_{rh}$ is now calculated as:

$$b_{rh\_new} = \frac{x_{+w} - x_{-w}}{c_w}$$

where $x_{+w}$ is the average of the measurements at concentration of the annual limit value in the presence of water vapour, in µg/m$^3$; $x_{-w}$ is the average of the measurements at concentration of the annual limit value in the absence of water vapour, in
µg/m$^3$; and $c_w$ is the water vapour concentration of the test (19 mmol/mol). Calculated parameter $b_{rh\_new}$ has to be $\leq 0.015$ µg/m$^3$/(mmol/mol). It is interesting to note that this test is not carried out anymore at 40 µg/m$^3$.

### 2.2.3.2 Second set of experiments

An in-depth study of the influence of humidity on measurements was carried out by measuring several reference mixtures of benzene in air (5 µg/m$^3$ nominal concentration) with different absolute humidity (AH) values ranging from 0 to 32 g/m$^3$. These
tests were repeated with a reference mixture of 40 µg/m$^3$ nominal concentration of benzene in air.

Humidity in the range 0-17 g/m$^3$ was obtained at 20 °C and relative humidity ranging from 0 to 99%. Higher absolute humidity was attained increasing the working temperature to 35 °C. Results in section 3.1 showed that sample temperature inside the tested range did not influence benzene measurements. Thus, this parameter can be changed in order to achieve a high absolute humidity in the samples.

### 3 Results and discussion

### 3.1 Influence of sample temperature on benzene measurements

As mentioned in Section 2.2.2, a test to evaluate the influence of sample temperature was carried out. The rationale for this was to know if this parameter affects the readings. If it is not the case, temperature can be changed during the tests and, therefore, the maximum absolute humidity tested is not limited by the saturation humidity of the sample at 20 °C. Table 1
shows the results of the tests when analyser I measured two reference gas mixtures (40 µg/m$^3$ and 5 µg/m$^3$ nominal

concentration benzene in air) subjected to two temperature cycles (20 ℃ / 5 ℃ / 20 ℃ and 20 ℃ / 35 ℃ / 20 ℃). Similar results were obtained for analyser II. As it can be seen, the change in sample temperature did not produce any significant change in readings and, thus, temperature was increased to 35 ℃ in some of the tests in order to work with a higher absolute humidity in our reference gas mixtures. The non-dependence of measurements on sample temperature can be explained by the

fact that the sample is heated initially in the oven to 50 ℃, the initial temperature of the sample being irrelevant in the whole process.

## 3.2 Compliance with the requirements of EN 14662-3:2005

Table 2 summarises the results obtained when carrying out the tests described in section 2.2.3.1. Whereas humidity does not have a significant influence on readings at 0.5 µg/m$^3$ level, it does at 40 µg/m$^3$ (calculated $b_{rh}$ coefficients of 18% and 21% for

analyser I and II, respectively). This is a negative influence, that is, readings are lower than expected when the relative humidity increases for a constant temperature. Moreover, calculated coefficient $b_{rh}$ for both anaslysers turned out to be higher than 4% (the maximum variation allowed in the Standard EN 14662-3:2005) when the tests were carried out with 40 µg/m$^3$ of benzene in air. In order to study deeper this phenomenon, the tests described in section 2.2.2.2 were carried out and the results are shown in section 3.3. Compliance of analyser I with the requirements of the new Standard was calculated from a set of results

in Table 3 (T= 20 ℃, relative humidity= 81%, $C_{ref}$= $C_{-w}$= 5.31 µg/m$^3$, $C_{meas}$=$C_{+w}$=4.99, $C_w$= 18.3 mmol/mol). A value of $b_{hr\_new}$ equal to 0.0175 µg/m$^3$/(mmol/mol) was obtained, which is higher than the new performance criterion (0.015 µg/m$^3$/(mmol/mol)). This interference would be even larger if the tests were conducted at 40 µg/m$^3$ (see Section 3.3). From our perspective, when type-approving GC-PID analysers, testing the humidity interference with a value in between the annual limit and the span (e.g. 20 µg/m$^3$) would be interesting to detect important biases at high benzene amount fractions, which can

be present, for instance, in industrial areas or close to petrol stations.

## 3.3 Influence of humidity on benzene measurements

Table 3 summarises the humidity conditions, the reference concentration of benzene generated (after considering the dilution effect of water vapour), the average reading of analyser I and the calculated relative difference from the reference concentration

of each test. These differences were plotted against the absolute humidity of the test, Figure 2.

There is a clear linear relationship between analyser readings and absolute humidity. For a given benzene amount fraction, the higher the absolute humidity in the sample the lower the chromatograph readings. This result was previously obtained by Barksy et al. (Barksy et al., 1985) but using concentrations of volatile organic compounds at ppm levels.

The data in Figure 2 were fitted by linear least-squares regression, which gave the following equations: $E$= -1.066·$AH$+4.783

(r$^2$=0.91) and $E$= -1.557·$AH$-3.341 (r$^2$=0.94) for nominal reference benzene concentrations of 5 µg/m$^3$ and 40 µg/m$^3$, respectively. $E$ is the relative difference between the reference concentration generated in the test chamber and the analyser reading. Differences between the slopes were studied to find out whether they were significantly different; for this, we used a

t-value calculated from t =(m$_1$-m$_2$)/SE(m$_1$-m$_2$), where m$_1$ and m$_2$ are the slopes of the two straight lines compared and SE(m$_1$-m$_2$) is the standard error of the difference, calculated as the square root of the quadratic sum of the standard error of each slope. A t-value of 2.272 was obtained. This value was higher than the critical one (2.145) for p=0.05 and 14 degrees of freedom (df=(n$_1$-2)+(n$_2$-2), where n$_1$ and n$_2$ are the number of data in the least-square regressions of Figure 2), which meant that the

difference in the slopes was significant and could not be attributed to random measurement error. This is interesting as it shows that the variation of readings by effect of ambient humidity is more pronounced at higher ambient ratios of benzene. Moreover, higher concentrations of benzene are more affected by ambient water vapour as for the same absolute humidity, relative differences are higher in the tests at 40 µg/m$^3$ than at 5 µg/m$^3$.

For 35 °C and 80% relative humidity (31 g/m$^3$ absolute humidity approx.) the bias in readings was 33% and 47% for a reference

concentration of benzene in air of 5 µg/m$^3$ and 40 µg/m$^3$, respectively. These conditions, although a bit extreme, can easily occur in many locations (e.g. Mediterranean areas in summer). Less extreme conditions can also have an important bias in readings (for instance, at 20 °C and 50% relative humidity there is a 2.2% bias in the concentration readings at 5 µg/m$^3$ level and 13% at 40 µg/m$^3$). Considering a location where mean annual benzene concentrations are close to the annual limit value (5 µg/m$^3$), a bias in measurements of approximately 2% can be easily expected due to a water vapour mixing ratio close to 8.6

g/cm$^3$. This bias can be acceptable, taking into consideration that benzene data quality objective in current legislation for fixed measurements is 25%. Thus, it should be incorporated to the uncertainty budget of the measurements with no need for further corrections. Moreover, if ambient concentrations are below the annual limit value, the interference of environmental humidity although not negligible will not change the air quality situation of that area. However, if benzene ambient ratios are above, measurements will be systematically underestimated by effect of ambient humidity, precisely in those areas where a stricter

control of concentrations is required. It could be the case of a location that apparently meets the air quality limits because concentrations are underestimated but, in reality, its environmental situation is not acceptable. Thus, it is in these cases where humidity interference on measurements should be addressed. Areas with concentrations of benzene above the annual limit value are widely reported in the literature (Anttila et al., 2016; Bruinen De Bruin et al., 2008; Licen et al., 2016; Al Madhoun et al., 2011).

In our tests, the baseline did not change when zero gas with different amount fractions of water vapour was measured. The peak shapes and the elution times did not change either when measuring a constant amount fraction of benzene with different amount fractions of humidity. This led us to think that water vapour does not interfere in the preconcentration and separation steps and it is the PID the part of the instrument most affected by the humidity.

The effect of humidity on PID performance has been proved to be double (MSA, 2005). Despite the ionising potential of water

vapour being higher than the energy of the PID, it can produce a small background signal at high non-condensing relative humidity, overestimating VOC concentrations. The second effect is the quenching of part of the UV light. When the analysers have been calibrated with dry gases and they measure a sample with humidity, the water vapour molecules in the sample absorb part of the UV radiation emitted. For a given concentration of benzene, the higher the absolute humidity in the sample the higher the absorption of UV radiation and the less energy available to ionise the molecules of benzene. This bias depends not

only on the water vapour concentration but also on the benzene one, as we have checked in our tests. For a given concentration of humidity, if the concentration of benzene is very low (e.g. 0.5 μg/m$^3$) the residual UV radiation, that is, the radiation not absorbed by the water vapour, is enough to ionise and, therefore, quantify all the molecules of benzene in the sample. This seems to be the case of the tests conducted in section 3.1 at 0.5 μg/m$^3$, as no effect was observed when changing the amount

fraction of water vapour in the reference mixture. However, as the amount fraction of benzene increases, the residual UV radiation may not be able to ionise all the molecules of benzene, as it is apparently happening with the samples with 5 μg/m$^3$ and 40 μg/m$^3$ benzene in air. From these two effects –background signal and radiation quenching-, the latter seems to be the most influencing as there is a decrease in readings with humidity and not an increase as it would be expected from a background signal effect.

A third phenomenon may be occurring as well. The benzene radical cation formed after ionisation of benzene can react with water to give hydroxycyclohexadienyl radical, which in turn can dissociate to benzene and OH radicals (Eberhardt, 1981). This effect is in line with the quenching effect of water vapour as both of them reduce the amount of ionised benzene reaching the electrodes.

There are a few works in the literature that use *in situ* GC-PID to measure benzene in air (Bruno et al., 2001; Kelessis et al.,

2006; Villanueva et al., 2012). Bruno et al. (2001) and Villanueva et al. (2012) use gas chromatographs from Syntech Spectras so a study of the influence of humidity in their measurements could be done as long as calibration details and relative humidity data are provided.

Bruno et al. (2001) mainly focused on source apportionment and they do not provide information related to ambient humidity. The average concentration of benzene during the sampling period was $4 \pm 1.6$ μg/m$^3$ so measurements were close to the annual

limit value but, mainly, below it, so around 2% bias is expected at 20 ºC and 50% relative humidity if calibration was carried out with dry gases. Villanueva et al. (2012) intercompared tropospheric ozone, benzene and toluene by a commercial DOAS and conventional monitoring techniques. The instrument used to measure benzene and toluene was the same as the one used in our work. They mention the use of certified gas mixtures to calibrate their instruments. There is no mention to water vapour in the mixtures so it is assumed that they are dry gases. Their results show that average levels for ozone, benzene and toluene

obtained with DOAS were higher than those obtained with UV photometry for ozone and GC-PID for the aromatics. The largest differences found are for benzene. Although the water interference found in our work is compatible with the results obtained in Villanueva et al. (2012), it may not explain the whole difference between the analytical techniques used, which is up to 50% in some cases. A quantification in this paper of the bias in their measurements by GC-PID due to water vapour is not possible due to lack of temperature and relative humidity data.

The influence of humidity on many air quality monitoring techniques has always been a major problem. PID detectors are not the only ones affected. FID were proved to be affected as well (LeBouf et al., 2013); however, these tests were performed at ppm levels. Among the reference measurement techniques to measure air pollutants, chemiluminescence with ozone to measure NO and NO$_2$ is also humidity-dependent (Gerboles et al., 2003; Hayden, 2003; Miñarro and Ferradás, 2012; Steinbacher et al., 2007); and also UV photometry to measure ozone (Wilson and Birks, 2006). Recently, Bluhme et al.

(Bluhme et al., 2016) have shown that measurements of $SH_2$ by UV fluorescence are also affected. The interference mechanism is different in each technique but the result is always an underestimation of measurements. Some manufacturers have opted for adding filters or driers to their equipment in order to keep humidity in the sample to a minimum. These implementations have been proved to reduce biases in some cases (Bluhme et al., 2016; Steinbacher et al., 2007; Wilson and Birks, 2006). An

alternative to scrubbers, which have the drawback of potentially adsorbing the target molecule, is calibration with wet gases. Ideally, calibration procedures should be done at the same ambient conditions as sampling. Calibration with wet gases may reduce measurement uncertainty due to environmental humidity in many cases. However, a thorough work regarding short and long-term stability of wet calibration gases in gas cylinders should be first tackled by metrology institutes. Using wet calibration gases obtained by dynamic dilution could bridge the gap and help reduce the uncertainty of benzene measurements

and other pollutants in ambient air.

The behaviour observed in this work is likely to be shown by GC-PID instruments by other manufacturers, although to a different extent, which means that benzene concentrations –and, presumably, given the nature of the interference, ethylbenzene, toluene and xylenes concentrations as well- may be systematically underestimated. In areas where ambient concentrations of benzene are usually above the annual limit value, the humidity interference on measurements should be

urgently addressed. A joint effort from manufacturers, metrology institutes and users is advisable to reduce the bias due to ambient humidity on BTEX measurements obtained by GC-PID –but also on measurements of other atmospheric pollutants-, as relievable data is the starting point for a correct environmental management.

**Conclusions**

In this work, the influence of sample temperature and ambient humidity on benzene measurements obtained with an automated

*in situ* GC-PID is studied. Sample temperature turned out not to influence measurements between 5 and 35 ℃. Regarding humidity, the chromatograph was calibrated with dry gases, which is nowadays a current practice, and, subsequently, different amount fractions of humidity were added to the reference mixture. The absolute humidity tested ranged from 0 to 31 $g/cm^3$. The dilution effect of adding water vapour was taken into account in the reference concentration calculation.

When measuring 5 $\mu g/m^3$ of benzene in air, biases in readings ranged from 1 to 32% depending on the absolute humidity in

the gas mixture. For an absolute humidity close to 8.6 $g/cm^3$ –corresponding to a relative humidity of 50% at 20 ℃- the bias in measurements is about 2.2%. Tests were repeated with a 40 $\mu g/m^3$ benzene in air mixture. In this case, biases of up to 47% were obtained when the absolute humidity in the sample was 30 $g/cm^3$. A less extreme absolute humidity in the sample (8 $g/cm^3$) produced a bias of approximately 13%. Results show that water vapour interference depends on both the water and benzene amount fractions in the sample.

If the concentrations of benzene in a certain location are far below the annual limit value (5 $\mu g/m^3$), the bias due to water interference can be acceptable, taking into consideration that benzene data quality objective in current legislation for fixed measurements is 25%. Thus, it should be incorporated to the uncertainty budget of the measurements with no need for further

corrections. Moreover, if ambient concentrations are below the annual limit value, the interference of environmental humidity although not negligible will not change the air quality situation of that area. However, if benzene ambient ratios are above, measurements will be systematically underestimated by effect of ambient humidity, precisely in those areas where a stricter control of concentrations is required. Thus, it is in these cases where humidity interference on measurements should be

addressed. Using appropriate scrubbers or wet calibration gases could help reduce measurement uncertainty of benzene and many other air pollutants monitored with analytical techniques also affected by water vapour.

**Acknowledgements**

We would like to acknowledge the Consejería de Agua, Agricultura y Medio Ambiente of the Comunidad Autónoma de la Región de Murcia for its financial support and for the facilities to carry out this work.

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

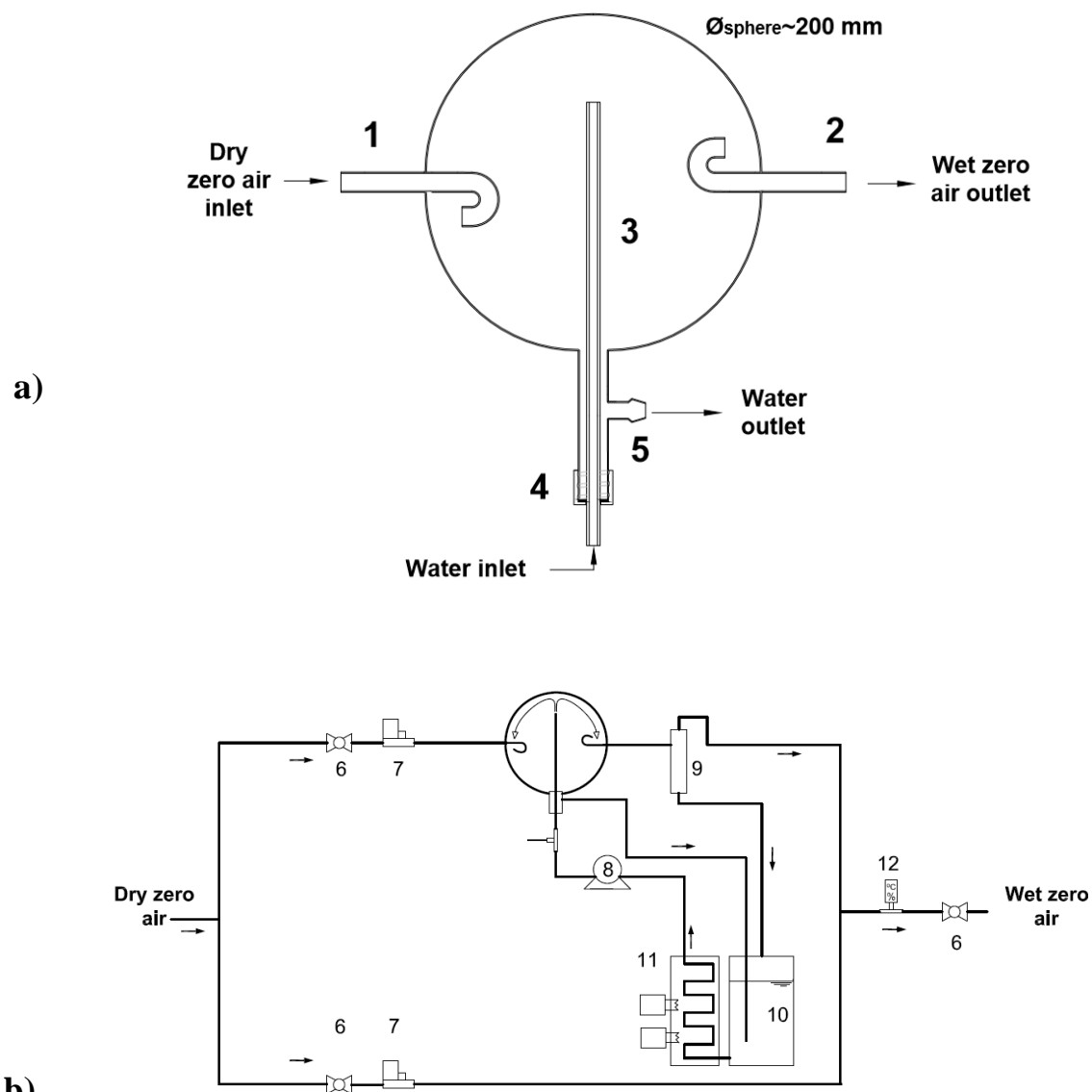

**Figure 1: Schematic of the (a) humidifier and (b) the elements that comprise the humidifying system used in this work. 1: dry zero air inlet, 2: wet zero air outlet, 3 and 4: water inlet, 5: water outlet, 6: shut-off valves, 7: mass flow controllers, 8: water pump, 9: cyclone, 10: water container, 11: thermostatic bath, 12: thermo-hygrometer.**

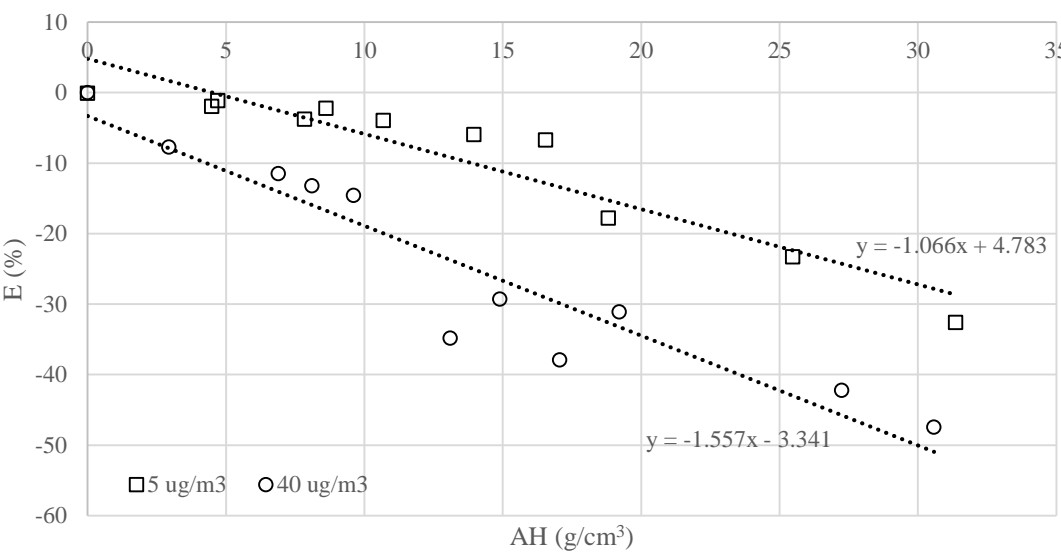

**Figure 2: Relative differences in readings from the reference value of concentration as a function of the absolute humidity of the sample.**

20

| Temperature cycle | 20 ºC | 5 ºC | 20 ºC |
|---|---|---|---|
| Average reading of concentration ($\mu g/m^3$) when measuring a benzene concentration of $39.96 \pm 0.30$ $\mu g/m^3$ | 39.96 (0.45) | 39.69 (0.26) | 39.82 (0.29) |
| Average reading of concentration ($\mu g/m^3$) when measuring a benzene concentration of $4.77 \pm 0.08$ $\mu g/m^3$ | 4.77 (0.08) | 4.76 (0.06) | 4.78 (0.06) |
| **Temperature cycle** | **20 ºC** | **35 ºC** | **20 ºC** |
| Average reading of concentration ($\mu g/m^3$) when measuring a benzene concentration of $39.96 \pm 0.30$ $\mu g/m^3$ | 39.82 (0.29) | 39.78 (0.50) | 39.87 (0.25) |
| Average reading of concentration ($\mu g/m^3$) when measuring a benzene concentration of $4.77 \pm 0.08$ $\mu g/m^3$ | 4.78 (0.06) | 4.77 (0.10) | 4.78 (0.11) |

**Table 1: Analyser I readings when subjected to changes in sample temperature. In brackets, the standard deviation of the measurements.**

| Test nominal concentration: 0.5 µg/m³ | Average value Analyser I (µg/m³) | Average value Analyser II (µg/m³) |
|---|---|---|
| 20% relative humidity | 0.53 (0.01) | 0.55 (0.01) |
| 80% relative humidity | 0.51 (0.01) | 0.53 (0.01) |
| $b_{rh}$ (%) | 3.1 | 3.9 |
| **Test nominal concentration: 40 µg/m³** | | |
| 20% relative humidity | 44.3 (0.17) | 46.7 (0.47) |
| 80% relative humidity | 36.9 (0.23) | 37.9 (0.18) |
| $b_{rh}$ (%) | 18.2 | 20.8 |
| **Test nominal concentration: 5 µg/m³** | | |
| 0% relative humidity | 5.31 (0.06) | |
| 80% relative humidity | 4.99 (0.07) | |
| $b_{rh\_new}$ (µg/m³/(mmol/mol)) | 0.0175 | |

**Table 2: Analyser readings when subjected to changes in ambient humidity for a constant reference concentration and sensitivity coefficients to humidity ($b_{rh}$ and $b_{rh\_new}$). Test temperature: 20 ± 2 ºC. In brackets, the standard deviation of the measurements.**

**T= 20 ± 2 ºC; $C_{std}$= 1053.5 ± 26.3 µg/m³; $q_z$= 11.636 ± 0.002 l/min; $q_{std}$=0.05999 ± 6.265·10⁻⁵ l/min**

| RH (%) | AH (g/cm³) | W (g H₂O/g dry air) | $q_{H2O}$ (l/min) | $q_t$ (l/min) | $C_{ref}$ (µg/m³) | $C_{meas}$ (µg/m³) | E (%) |
|---|---|---|---|---|---|---|---|
| 0.4 | 0.0689 | 5.74·10⁻⁵ | 1.069·10⁻³ | 11.697 | 5.40 | 5.40 | 0.00 |
| 26 | 4.480 | 0.00373 | 0.0695 | 11.765 | 5.37 | 5.27 | -1.95 |
| 50 | 8.616 | 0.00718 | 0.134 | 11.830 | 5.34 | 5.23 | -2.20 |
| 62 | 10.68 | 0.00890 | 0.166 | 11.862 | 5.33 | 5.12 | -3.95 |
| 81 | 13.96 | 0.0116 | 0.217 | 11.913 | 5.31 | 4.99 | -5.94 |
| 96 | 16.54 | 0.0138 | 0.257 | 11.953 | 5.29 | 4.93 | -6.70 |

**T= 35 ± 2 ºC; $C_{std}$= 1053.5 ± 26.3 µg/m³; $q_z$= 11.637 ± 0.002 l/min; $q_{std}$=0.05999 ± 6.265·10⁻⁵ l/min**

| RH (%) | AH (g/cm³) | W (g H₂O/g dry air) | $q_{H2O}$ (l/min) | $q_t$ (l/min) | $C_{ref}$ (µg/m³) | $C_{meas}$ (µg/m³) | E (%) |
|---|---|---|---|---|---|---|---|
| 0.2 | 0.0345 | 2.871·10⁻⁵ | 5.347·10⁻⁴ | 11.697 | 5.40 | 5.40 | 0.00 |
| 12 | 4.70 | 0.00392 | 0.0730 | 11.769 | 5.37 | 5.31 | -1.12 |
| 20 | 7.84 | 0.00653 | 0.122 | 11.818 | 5.35 | 5.15 | -3.75 |
| 48 | 18.82 | 0.0157 | 0.292 | 11.988 | 5.27 | 4.34 | -17.8 |
| 65 | 25.48 | 0.0212 | 0.395 | 12.091 | 5.23 | 4.01 | -23.3 |
| 80 | 31.36 | 0.0261 | 0.487 | 12.183 | 5.19 | 3.50 | -32.6 |

**T= 25 ± 2 ºC; $C_{std}$= 1053.5 ± 26.3 µg/m³; $q_z$= 9.099 ± 0.002 l/min; $q_{std}$=0.3599 ± 4.125·10⁻⁴ l/min**

| RH (%) | AH (g/cm³) | W (g H₂O/g dry air) | $q_{H2O}$ (l/min) | $q_t$ (l/min) | $C_{ref}$ (µg/m³) | $C_{meas}$ (µg/m³) | E (%) |
|---|---|---|---|---|---|---|---|
| 1.1 | 0.190 | 1.579·10⁻⁴ | 2.299·10⁻³ | 9.461 | 40.07 | 40.08 | 0.02 |
| 17 | 2.929 | 0.00244 | 0.0355 | 9.494 | 39.93 | 36.85 | -7.72 |
| 40 | 6.893 | 0.00574 | 0.0836 | 9.543 | 39.73 | 35.18 | -11.46 |
| 47 | 8.099 | 0.00675 | 0.0983 | 9.557 | 39.67 | 34.43 | -13.21 |
| 76 | 13.10 | 0.0109 | 0.1589 | 9.618 | 39.42 | 25.70 | -34.80 |
| 99 | 17.06 | 0.0142 | 0.207 | 9.666 | 39.23 | 24.37 | -37.87 |

**T= 35 ± 2 ºC; $C_{std}$= 1053.5 ± 26.3 µg/m³; $q_z$= 9.098 ± 0.002 l/min; $q_{std}$=0.3599 ± 4.125·10⁻⁴ l/min**

| RH (%) | AH (g/cm$^3$) | W (g H$_2$O/g dry air) | $q_{H2O}$ (l/min) | $q_t$ (l/min) | $C_{ref}$ (µg/m$^3$) | $C_{meas}$ (µg/m$^3$) | E (%) |
|---|---|---|---|---|---|---|---|
| 1.8 | 0.706 | $5.88 \cdot 10^{-3}$ | $8.56 \cdot 10^{-3}$ | 9.466 | 40.05 | 40.09 | 0.10 |
| 25 | 9.604 | 0.0080 | 0.117 | 9.574 | 39.60 | 33.83 | -14.56 |
| 38 | 14.90 | 0.0125 | 0.182 | 9.640 | 39.33 | 27.82 | -29.28 |
| 49 | 19.21 | 0.0161 | 0.235 | 9.693 | 39.12 | 26.95 | -31.10 |
| 70 | 27.24 | 0.0229 | 0.333 | 9.791 | 38.72 | 22.38 | -42.20 |
| 78 | 30.57 | 0.0257 | 0.374 | 9.832 | 38.56 | 20.28 | -47.41 |

**Table 3: Relative (RH) and absolute (AH) humidity, mass fraction of water vapour in the dry air (W), reference concentration of benzene ($C_{ref}$), average reading of analyser I ($C_{meas}$) and relative difference between measurements and reference values (E). $q_t$ is the total flow rate, calculated as the sum of the flowrate of zero air ($q_z$), water vapour ($q_{H2O}$) and the flow rate from the benzene reference gas mixture ($q_{std}$). $C_{std}$ is the amount fraction of benzene in the gas cylinder.**