# Peer review of "Influence of sample temperature and environmental humidity on measurements of benzene in ambient air by transportable GC-PID"

_Atmospheric Measurement Techniques, 2017_

## Referee Comment (RC1) · Anonymous Referee #1 · 2 Jun 2017

**General comments.**

The manuscript by Romero-Trigueros et al. address an important issue regarding air quality monitoring of benzene by transportable GC-PID. The tests are reported clearly and the implications discussed appropriately. However, my main concerns about the manuscript are:

1) Focusing on benzene: It would be more relevant to have investigated all the species analysed by this specific GC-PID, often referred to as "BTEX analyser" (for benzene, toluene, ehtylbenzene, and xylenes).

[Figure]

2) Previously reported influence of pressure: A previous publication form the same authors in a different journal reported that data from this type of GC-PID might be affected by pressure difference.

These issues hint at a disputable publication strategy spreading results of the characterization of this monitor in several manuscript, increasing the publication count of the authors, but diluting the relevant information for the users of this type of monitors. Therefore, I would recommend to reject this manuscript.

**Specific comments.**

- The content of the manuscript is well reflect in the abstract but it has been omitted that temperature influence has also been tested. Even though the result is that the temperature has no influence, it might be worth to emphasize this result in the abstract (or even in the title of the manuscript).

- page 3, line 25-26: Tests with temperature should have their dedicated experimental section and the results should be reported only in section 3.2.

- page 4, line 7: It should be clarified (possibly in the introduction) that the EN Standard the authors are referring to citation "EU, 2008".

- page 5, line 7: The sentence starting with "This influence has a negative sign,..." should be revised as the wording seems odd.

- page 6, line 4-5: Statistics have been used but are poorly described in this one sentence. What is the value for p=0.05 and 14 degrees of freedom and what are n1 and n2? This should be improved and clarified for readers that are not familiar with such statistic tools.

- page 6, line 20-24: It could be discussed in a bit more detailed how the results

reported in the literature might have suffered from the presented influence of RH, given the reported environmental conditions of the measurements.

- from page 6, line 25: This whole paragraph is a bit strange to me. It is not clear if the discussion is related to findings (or speculation?) of the authors or from the literature (in which case, references would be missing). The paragraph starts with "have been proved", but then rest reads like possible explanation for the observed influence of humidity on the results. Moreoever, if the effect of water on preconcentration and chromatographic separation is not discussed (e.g. baseline, peak shapes, etc.) because is thought not to be relevant, this should be briefly motivated. Do authors expect the water concentration at the detector when benzene elutes to be the same as the ambient water concentration during sampling?

- page 7, line 25: If "presumably" TEX are also affected, authors should at least mention briefly why they did not include them in their present work and if they are planning to do it in the near future. If they do have results for these compounds, they should not be withdrawn and included in the present manuscript.

**Technical corrections**

- page 1, line 25: typo "h0ematologic"

- page 2, line 28: Trigueros et al., 2016 is missing in the bibliography

- page 3, line 12-13: use either commas or long dash (–) to separate "the ratio of the actual mass of water vapour present in the sample to the mass of the dry air"

- page 4, line 5: replace "pressure" with "humidity"

- page 4, line 20: I suggest to use "with different absolute humidity (AH) values"

- page 5, line 20: Word order: "the initial temperature of the sample being irrelevant in

the whole process"

- page 9, line 24: This is not a DOI, this is an URL.

- page 9, line 32: Remove "http://dx.doi.org/".

---

## Referee Comment (RC2) · Anonymous Referee #2 · 22 Jun 2017

General Comments The manuscript provides the influence of ambient humidity on benzene measurements carried out with an automated in situ GC-PID that is the instrument usually used in air quality monitoring networks. I consider that is an interesting study and therefore, I recommend the manuscript for publication in Atmospheric measurement Techniques. Some specific comments are given below to be clarified by authors before publication.

Specific Comments Pag. 2 27. What is the brand, model of the dynamic dilution system? Please include in the text. Pag. 3, line 18. Is the mixture benzene in nitrogen a SRM, if so please indicate it. Who is the supplier of the gas cylinder? Which is

[Figure]

the purity? Please, indicate it in the text. Are the flow meters calibrated periodically? How? Pag. 4, line 10. Authors refer to Standard EN 14662-3:2005 in order to calculate the coefficient brh to see the influence of relative humidity but this standard is already cancelled and the equation 2 does not appear in the EN 14662-3:2015. Why do authors think that this equation has been removed from the updated Standard? If the relative humidity is so important at high concentration of benzene as demonstrated in the manuscript why it has been deleted? Table 1. if (0.01) is the standard deviation, please indicate it in the table. Table 3 . A "(" should be before $\mu$g/m3) in Cmeas. What is qt? in the text are defined qz and qH2O but not qt.

On the other hand, have the authors checked the memory effect when using high concentration of benzene (40 ug/m3)?

Technical comments

Pag. 1, line 25: revise "h0ematologic" Pag. 1 , line 29. Reference properly the Standard EN in reference section. The standard number is missing. Pag. 4. , line 7 and pag. 5, line 4. I think that authors should change "2005" by "2015". Pag. 4, line 18. Section 2.2.2.1 should be 2.2.2.2 Pag. 5, line 6. Revise the sentence: Moreover, coefficient. . .. In air.
* * *

---

## Author Comment (AC1) · 28 Jun 2017

General comments.

The manuscript by Romero-Trigueros et al. address an important issue regarding air quality monitoring of benzene by transportable GC-PID. The tests are reported clearly and the implications discussed appropriately. However, my main concerns about the manuscript are:

1) Focusing on benzene: It would be more relevant to have investigated all the species analysed by this specific GC-PID, often referred to as "BTEX analyser" (for benzene,

toluene, ehtylbenzene, and xylenes).

Focusing on benzene is supported by the fact that this is the only VOC in the European Union with limit values and a standardized method for measuring its concentration in air. It is true that adding toluene, ethylbenzene and xylenes to our work would have improved the paper but these species were out of our goal. Measuring toluene, ethylbenzene and xylenes would have meant either repeating the tests for these species –increasing the project cost (more reference gases for each of them) and time; or having worked with a reference gas mixture with all these components, which would have increased the uncertainty of the mixture composition and its stability. For all of this and, mainly, because benzene was our target VOC, we only carried out the tests with benzene. Nevertheless, in the future, the rest of compounds could be further investigated.

2) Previously reported influence of pressure: A previous publication from the same authors in a different journal reported that data from this type of GC-PID might be affected by pressure difference. These issues hint at a disputable publication strategy spreading results of the characterization of this monitor in several manuscripts, increasing the publication count of the authors, but diluting the relevant information for the users of this type of monitors. Therefore, I would recommend to reject this manuscript.

We the authors think that Reviewer #1 has been a bit harsh with us. It is true that we have published another paper related to the influence of pressure on GC-PID benzene measurements. These two papers, together with many more tests that have not been published, are the results of a PhD thesis. We believe that those tests with interesting results should be published independently in order to be treated deeper. If all the tests were to be included and discussed in a single publication, its length would have been too long. This is, of course, our point of view and does not have to be shared by everybody. But, again, we do not think this is enough to reject our paper as the results in it have not been published anywhere else.

Specific comments.

- The content of the manuscript is well reflected in the abstract but it has been omitted that temperature influence has also been tested. Even though the result is that the temperature has no influence, it might be worth to emphasize this result in the abstract (or even in the title of the manuscript).

The reviewer is right and temperature influence should be mentioned in the abstract and could be also mentioned in the title of the manuscript.

- page 3, line 25-26: Tests with temperature should have their dedicated experimental section and the results should be reported only in section 3.2.

Temperature tests could be treated as a main goal of the manuscript together with humidity (accordingly to previous comment) and, therefore, the manuscript should change accordingly. This would mean adding a new sub-section in the experimental.

- page 4, line 7: It should be clarified (possibly in the introduction) that the EN Standard the authors are referring to citation "EU, 2008".

"EU, 2008" refers to Directive 2008/50/EC. "EN, 2015" refers to EN Standard 14662-3:2015 and this will be stated clearer in the introduction. However, the preliminary test in section 3.1 was carried out according to EN Standard 14662-3:2015, which is the previous version of the standard. This is because these tests were carried out before the release of the new version. Even though the tests measuring the humidity influence have changed in the new version of the Standard, this is not important as those tests only revealed a significant influence that was subsequently studied deeper by designing new experiments. In other words, for us, the Standard was only the starting point which revealed a significant influence of ambient humidity. From there, we carried out a deeper study of the influence of humidity in measurements.

- page 5, line 7: The sentence starting with "This influence has a negative sign,..." should be revised as the wording seems odd.

The wording will be changed to "This is a negative influence, that is, . . ."

- page 6, line 4-5: Statistics have been used but are poorly described in this one sentence. What is the value for p=0.05 and 14 degrees of freedom and what are n1 and n2? This should be improved and clarified for readers that are not familiar with such statistic tools.

The information missing will be added (t-value for p=0.05 and 14 degrees of freedom is 2.145 and n1 and n2 are the number of data in the least-square regressions of Figure 2 (for nominal benzene concentrations of 5 and 40 $\mu$g/m3, respectively). Also, a mistake was detected. Our calculated t-value is higher than the tabulated one which means that we reject the null hypothesis, that is, there are significant differences in the two slopes not attributed to random errors. This will be corrected in the final manuscript.

- page 6, line 20-24: It could be discussed in a bit more detailed how the results reported in the literature might have suffered from the presented influence of RH, given the reported environmental conditions of the measurements.

A thorough revision of literature has been carried out in order to find works where benzene was measured with a similar instrument to that used in this work. Many works use active or passive sampling followed by ex situ thermal desorption and GC-PID or GC-FID (e.g. Marc et al., 2016; Fracasso et al. (2010); Allou et al. 2008; Tran et al., 2000). There are a few of them that use in situ GC-FID (e.g. Zhang et al. 2017; Durana et al. 2006) and in situ GC-PID (e.g. Bruno et al., 2001; Kelessis et al., 2006; Villanueva et al., 2012). Bruno et al. (2001) and Villanueva et al. (2012) use gas chromatographs from Syntech Spectras so a study of the influence of humidity in their measurements could be done as long as calibration details and relative humidity data are provided. Bruno et al. (2001) mainly focused on source apportionment and they do not provide information related to ambient humidity. The average concentration of benzene during the sampling period was 4 $\pm$ 1.6 $\mu$g/m3 so measurements were close to the annual limit value but, mainly, below it, so around 2% bias is expected at 20 °C and 50% relative humidity if calibration was carried out with dry gases.

[Figure]

Villanueva et al. (2012) intercompared tropospheric ozone, benzene and toluene by a commercial DOAS and conventional monitoring techniques. The instrument used to measure benzene and toluene was the same as the one used in our work. They mention the use of certified gas mixtures to calibrate their instruments. There is no mention to water vapour in the mixtures so it is assumed that they are dry gases. Their results show that average levels for ozone, benzene and toluene obtained with DOAS were higher than those obtained with UV photometry for ozone and GC-PID for the aromatics. The largest differences found are for benzene. Although the water interference found in our work is compatible with the results obtained in Villanueva et al. (2012), it does not explain the whole difference between the analytical techniques used.

These discussions could be added in the corrected manuscript.

- from page 6, line 25: This whole paragraph is a bit strange to me. It is not clear if the discussion is related to findings (or speculation?) of the authors or from the literature (in which case, references would be missing). The paragraph starts with "have been proved", but then rest reads like possible explanation for the observed influence of humidity on the results. Moreoever, if the effect of water on preconcentration and chromatographic separation is not discussed (e.g. baseline, peak shapes, etc.) because is thought not to be relevant, this should be briefly motivated. Do authors expect the water concentration at the detector when benzene elutes to be the same as the ambient water concentration during sampling?

There was a missing reference that has been added (MSA, 2005). The technical document can be found in http://media.msanet.com/NA/USA/PermanentInstruments/GasSensorsTransmitters/SaveToxSafeVOC/07-2092WhitePaperPID.pdf

The baseline did not change when zero gas with different amount fractions of water vapour was measured. The peak shapes did not change either when measuring a

constant amount fraction of benzene with different amount fractions of humidity. This is why we think that water vapour does not interfere in the preconcentration and separation steps. This will be included in the final manuscript.

Although the amount fraction of water vapour was not measured at the outlet of the instrument, it is assumed to be the same as the inlet, because this instrument does not have any dryer or scrubber.

- page 7, line 25: If "presumably" TEX are also affected, authors should at least mention briefly why they did not include them in their present work and if they are planning to do it in the near future. If they do have results for these compounds, they should not be withdrawn and included in the present manuscript.

As mentioned previously, these species were out of the scope of this work. We only studied benzene as it is the only VOC with limit values in air quality in the European Union. This will be stated clearer in the manuscript.

Technical corrections

- page 1, line 25: typo "h0ematologic" Thanks. This will be changed in the corrected manuscript.

- page 2, line 28: Trigueros et al., 2016 is missing in the bibliography It appears as Romero-Trigueros et al. and, therefore, it will be changed in the text.

- page 3, line 12-13: use either commas or long dash (–) to separate "the ratio of the actual mass of water vapour present in the sample to the mass of the dry air" The comma will be removed.

- page 4, line 5: replace "pressure" with "humidity" Thanks. This will be changed

- page 4, line 20: I suggest to use "with different absolute humidity (AH) values" The word "values" will be added to the sentence. Thank you.

- page 5, line 20: Word order: "the initial temperature of the sample being irrelevant in

the whole process" Thank you for the suggestion.

- page 9, line 24: This is not a DOI, this is an URL. Thanks for this.

- page 9, line 32: Remove "http://dx.doi.org/". Thanks.

———————————————————

---

## Referee Comment (RC3) · Anonymous Referee #2 · 11 Jul 2017

I think that authors should include in the introduction information more relevant to support their results such as the work of Villanueva et al., 2012 that they cite in the letter to referee 1. GC-PID (the same model of the study, Syntech spectras GC 955) is usually used in the monitoring stations at least in Castilla La Mancha region. Puertollano is an industrialized area with levels of benzene higher than in other areas, therefore it is very important to know If the measurements should be corrected for RH because they could be underestimated.

---

## Author Comment (AC2) · 13 Jul 2017

Response to Reviewer #2

General Comments

The manuscript provides the influence of ambient humidity on benzene measurements carried out with an automated in situ GC-PID that is the instrument usually used in air quality monitoring networks. I consider that is an interesting study and therefore, I recommend the manuscript for publication in Atmospheric Measurement Techniques. Some specific comments are given below to be clarified by authors before publication.

[Figure]

Thank you for this and your valuable suggestions.

Specific Comments

Pag. 2 27. What is the brand, model of the dynamic dilution system? Please include in the text.

Our dynamic dilution system was built in-house. This will be clarified in the text. All the description in the text regarding the humidifier, the flow rate and temperature control refers to this in-house built system. Some of its parts (such as the humidifier) were designed by the authors and some of them were bought (such as the mass flow controllers).

Pag. 3, line 18. Is the mixture benzene in nitrogen a SRM, if so please indicate it. Who is the supplier of the gas cylinder? Which is the purity? Please, indicate it in the text.

The mixture of benzene in nitrogen was provided by Abelló Linde. Its concentration was certified by Linde according to the Standard ISO 6141:2007. The expanded uncertainty of the amount fraction of benzene in the mixture was 5%. This information will be added to the manuscript.

Are the flow meters calibrated periodically? How?

The mass flow controllers are at least annually calibrated against a Gilian Gilibrator (a NIOSH primary standard air flow calibrator) available in our lab. If the deviation of the MFC measurements is higher than 1% then the measurements are corrected with a calibration line. If lower, this bias is accepted and accounted for in the uncertainty assessment of the concentration of the gas mixtures.

Pag. 4, line 10. Authors refer to Standard EN 14662-3:2005 in order to calculate the coefficient brh to see the influence of relative humidity but this standard is already cancelled and the equation 2 does not appear in the EN 14662-3:2015. Why do authors think that this equation has been removed from the updated Standard?

[Figure]

We think that tests were likely changed because of the non-conformities of analysers with the Standard criterion for the span concentration. It is true that the span concentration (40 $\mu$g/m3) is quite high and such high levels are not usually found in the ambient air if measured away from the emission sources. However, from our point of view, it is interesting to test other concentrations apart from the limit value. From our perspective, testing this influence with a value in between the annual limit and the span (around 20 $\mu$g/m3) would be interesting to detect important biases at high benzene amount fractions, which can be present, for instance, in industrial areas.

If the relative humidity is so important at high concentration of benzene as demonstrated in the manuscript, why it has been deleted?

I am afraid we cannot answer this question. We guess that manufacturers may have been having issues with passing this test and, arguing that ambient concentrations of benzene are not usually as high as 40 $\mu$g/m3, they may have put pressure on normalizers to soften the performance criterion.

Table 1. if (0.01) is the standard deviation, please indicate it in the table.

It is the standard deviation. This information will be added in Table 1.

Table 3 . A "(" should be before $\mu$g/m3) in Cmeas. What is qt? in the text are defined qz and qH2O but not qt.

qt is the total flow rate, that is, the sum of qz, qH2O and the flow rate from the reference gas mixture. This will be specified in the caption of Table 3.

On the other hand, have the authors checked the memory effect when using high concentration of benzene (40 ug/m3)?

Indeed, we did. We followed standard EN 14662-3:2006 for this test (standard EN 14662-3:2016 does not substantially change this test). We measured a nominal concentration of benzene in air of 45 $\mu$g/m3 and then we switched to zero air. We carried out this test for Analyser I and II. For Analyser I we obtained the following 3 sequential measurement results: 46.59 $\mu$g/m3, -0.01 $\mu$g/m3 and -0.01 $\mu$g/m3. For Analyser II we obtained: 45.05 $\mu$g/m3, 0.32 $\mu$g/m3 and 0.31 $\mu$g/m3. Both analysers met the performance criteria of the Standard for this test (a measurement < 20% of the annual limit value for analysis n.2 (the one right after changing the concentration to zero) and <10% of the annual limit value for analysis n.3). Analyser II measured 0.05 $\mu$g/m3 in a fourth analysis and 0.01 $\mu$g/m3 in a fifth one, so we can conclude that this effect is not significant in this type of instruments. In any case, every time we changed the concentration of benzene we excluded the first measurement from the data analysis.

Technical comments

Pag. 1, line 25: revise "h0ematologic"

Thanks for spotting this.

Pag. 1 , line 29. Reference properly the Standard EN in reference section. The standard number is missing.

We will add the number of the standard in the text.

Pag. 4. , line 7 and pag. 5, line 4. I think that authors should change "2005" by "2015".

We will explain that the tests were carried out according to Standard EN 14662-3:2005 as these were carried out before the publication of the new standard. We will also describe the differences between the two standards regarding the humidity influence.

Pag. 4, line 18. Section 2.2.2.1 should be 2.2.2.2

Thanks for spotting this.

Pag. 5, line 6. Revise the sentence: Moreover, coefficient. . .. In air.

This sentence has been reworded to "Moreover, calculated coefficients brh for both analysers turned out to be higher than 4% (the maximum variation allowed in the Standard EN 14662-3:2005) when the tests were carried out with 40 $\mu$g/m3 of benzene in

air". This will be accordingly changed in the manuscript.

---

## Author Comment (AC3) · 13 Jul 2017

We will include a discussion of the paper by Villanueva et al in the final manuscript . We can cite this work in the introduction and then discuss how the results can be affected in the results and discussion section.

---

## Author Response (AR3)

INITIAL REVISION

**RESPONSE TO REVIEWER #1**

We appreciate the comments and the revision carried out by Reviewer #1 and we understand his/her concerns. We would like to kindly ask him/her to review our paper again, after incorporating his/her suggestions. Thank you very much.

**General comments.**

**The manuscript by Romero-Trigueros et al. address an important issue regarding air quality monitoring of benzene by transportable GC-PID. The tests are reported clearly and the implications discussed appropriately. However, my main concerns about the manuscript are:**

**1) Focusing on benzene: It would be more relevant to have investigated all the species analysed by this specific GC-PID, often referred to as "BTEX analyser" (for benzene, toluene, ehtylbenzene, and xylenes).**

Focusing on benzene is supported by the fact that this is the only VOC in the European Union with limit values and a standardized method for measuring its concentration in air. It is true that adding toluene, ethylbenzene and xylenes to our work would have improved the paper but these species were out of our goal. Measuring toluene, ethylbenzene and xylenes would have meant either repeating the tests for these species – increasing the project cost (more reference gases for each of them) and time; or having worked with a reference gas mixture with all these components, which would have increased the uncertainty of the mixture composition and its stability.

For all of this and, mainly, because benzene was our target VOC, we only carried out the tests with benzene. Nevertheless, in the future, the rest of compounds could be further investigated.

**2) Previously reported influence of pressure: A previous publication from the same authors in a different journal reported that data from this type of GC-PID might be affected by pressure difference. These issues hint at a disputable publication strategy spreading results of the characterization of this monitor in several manuscripts, increasing the publication count of the authors, but diluting the relevant information for the users of this type of monitors. Therefore, I would recommend to reject this manuscript.**

It is true that we have published another paper related to the influence of pressure on GC-PID benzene measurements. These two papers, together with many more tests that have not been published, are the results of a PhD thesis. We believe that those tests with interesting results should be published independently in order to be treated deeper. If all the tests were to be included and discussed in a single publication, its length would have been too long.

A paper similar to ours, dealing with water vapour interference in $H_2S$ monitors, was recently published in AMT journal and it was considered to be sufficiently comprehensive for this. This paper can be read here:

https://www.atmos-meas-tech.net/9/2669/2016/amt-9-2669-2016.html

For all of this, we would like to kindly ask Reviewer #1 to review the corrected manuscript again and consider it for publication. We really appreciate it.

**Specific comments.**

**- The content of the manuscript is well reflected in the abstract but it has been omitted that temperature influence has also been tested. Even though the result is that the temperature has no influence, it might be**
5 **worth to emphasize this result in the abstract (or even in the title of the manuscript).**

The reviewer is right and temperature influence should be mentioned in the abstract and be also mentioned in the title of the manuscript. These have been done. Please refer to the new title and abstract.

**- page 3, line 25-26: Tests with temperature should have their dedicated experimental section and the results should be reported only in section 3.2.**

10 Temperature tests are now treated as a main goal of the manuscript together with humidity (accordingly to previous comment) and, therefore, the manuscript has changed accordingly. Sections 2.2.2 and 3.1 are now devoted to the influence of sample temperature.

**- page 4, line 7: It should be clarified (possibly in the introduction) that the EN Standard the authors are**
15 **referring to citation "EU, 2008".**

"EU, 2008" refers to Directive 2008/50/EC. "EN, 2015" refers to EN Standard 14662-3:2015 and this has been stated clearer in the introduction (line 27 in page 1 and line 5 in page 2, respectively). However, the preliminary test in section 3.2 was carried out according to EN Standard 14662-3:2005, which is the previous version of the
20 standard. This is because these tests were carried out before the release of the new version. This is all explained in the new section 2.2.3.1. The water interference test of the new version is also explained in this section (line 7 in page 5).

Luckily enough, from the further tests that we performed in section 2.2.3.2 we were able to calculate the new
25 parameter $b_{rh}$, and these results have been included in section 3.2 (line 14, page 6).

**- page 5, line 7: The sentence starting with "This influence has a negative sign,..." should be revised as the wording seems odd.**

The wording has been changed to "This is a negative influence, that is, …" (line 10, page 6)

**- page 6, line 4-5: Statistics have been used but are poorly described in this one sentence. What is the value**
30 **for p=0.05 and 14 degrees of freedom and what are n1 and n2? This should be improved and clarified for readers that are not familiar with such statistic tools.**

The information missing has been added (t-value for $p=0.05$ and 14 degrees of freedom is 2.145 and $n_1$ and $n_2$ are the number of data in the least-square regressions of Figure 2 (for nominal benzene concentrations of 5 and 40 $\mu g/m^3$, respectively). Also, a mistake was detected. Our calculated t-value is higher than the tabulated one, which
35 means that we reject the null hypothesis, that is, there are significant differences in the two slopes not attributed to random errors. (Line 2, page 7).

**- page 6, line 20-24: It could be discussed in a bit more detailed how the results reported in the literature might have suffered from the presented influence of RH, given the reported environmental conditions of the measurements.**

A thorough revision of literature has been carried out in order to find works where benzene was measured with a similar instrument to that used in this work. Many works use active or passive sampling followed by *ex situ* thermal desorption and GC-PID or GC-FID (e.g. Marc et al., 2016; Fracasso et al. (2010); Allou et al. 2008; Tran et al., 2000). There are a few of them that use *in situ* GC-FID (e.g. Zhang et al. 2017; Durana et al. 2006) and *in situ* GC-PID (e.g. Bruno et al., 2001; Kelessis et al., 2006; Villanueva et al., 2012). Bruno et al. (2001) and Villanueva et al. (2012) use gas chromatographs from Syntech Spectras so a study of the influence of humidity in their measurements could be done as long as calibration details and relative humidity data are provided.

Bruno et al. (2001) mainly focused on source apportionment and they do not provide information related to ambient humidity. The average concentration of benzene during the sampling period was $4 \pm 1.6$ µg/m$^3$ so measurements were close to the annual limit value but, mainly, below it, so around 2% bias is expected at 20 ºC and 50% relative humidity if calibration was carried out with dry gases.

Villanueva et al. (2012) intercompared tropospheric ozone, benzene and toluene by a commercial DOAS and conventional monitoring techniques. The instrument used to measure benzene and toluene was the same as the one used in our work. They mention the use of certified gas mixtures to calibrate their instruments. There is no mention to water vapour in the mixtures so it is assumed that they are dry gases. Their results show that average levels for ozone, benzene and toluene obtained with DOAS were higher than those obtained with UV photometry for ozone and GC-PID for the aromatics. The largest differences found are for benzene. Although the water interference found in our work is compatible with the results obtained in Villanueva et al. (2012), we would need to know the pair of values temperature and relative humidity to assess the extend of the water vapour interference.

These discussions have been added in the corrected manuscript (from line 13 on in page 8).

**- from page 6, line 25: This whole paragraph is a bit strange to me. It is not clear if the discussion is related to findings (or speculation?) of the authors or from the literature (in which case, references would be missing). The paragraph starts with "have been proved", but then rest reads like possible explanation for the observed influence of humidity on the results. Moreoever, if the effect of water on preconcentration and chromatographic separation is not discussed (e.g. baseline, peak shapes, etc.) because is thought not to be relevant, this should be briefly motivated. Do authors expect the water concentration at the detector when benzene elutes to be the same as the ambient water concentration during sampling?**

There was a missing reference that has been added (MSA, 2005) in line 28, page 7. The technical document can be found in
http://media.msanet.com/NA/USA/PermanentInstruments/GasSensorsTransmitters/SaveToxSafeVOC/07-2092WhitePaperPID.pdf

The baseline did not change when zero gas with different amount fractions of water vapour was measured. The peak shapes did not change either when measuring a constant amount fraction of benzene with different amount fractions of humidity. This is why we think that water vapour does not interfere in the preconcentration and separation steps. This has been included in the final manuscript (line 24, page 7)

Although the amount fraction of water vapour was not measured at the outlet of the instrument, it is assumed to be the same as the inlet, because this instrument does not have any dryer or scrubber.

**- page 7, line 25: If "presumably" TEX are also affected, authors should at least mention briefly why they did not include them in their present work and if they are planning to do it in the near future. If they do have results for these compounds, they should not be withdrawn and included in the present manuscript.**

As mentioned previously, these species were out of the scope of this work. We only studied benzene as it is the only VOC with limit values in air quality in the European Union. This has been stated clearer in the manuscript (line 24, page 2).

**Technical corrections**

**- page 1, line 25: typo "h0ematologic"**
Thanks. This has been changed in the corrected manuscript (line 1, page 2).

**- page 2, line 28: Trigueros et al., 2016 is missing in the bibliography**
It appears now as Romero-Trigueros et al. and, therefore, it has been changed accordingly in the text (line 22, page 2 and line 5 page 3).

**- page 3, line 12-13: use either commas or long dash (–) to separate "the ratio of the actual mass of water vapour present in the sample to the mass of the dry air"**
The comma has been removed (line 18, page 3).

**- page 4, line 5: replace "pressure" with "humidity"**
Thanks. This has been changed in section 2.2.2.

**- page 4, line 20: I suggest to use "with different absolute humidity (AH) values"**
The word "values" has been added to the sentence. Thank you. (line 19, page 5).

**- page 5, line 20: Word order: "the initial temperature of the sample being irrelevant in the whole process"**
Thank you for the suggestion. (line 5, page 6)

**- page 9, line 24: This is not a DOI, this is an URL.**
Thanks for this. (line 12, page 11)

**- page 9, line 32: Remove "http://dx.doi.org/".**
Thanks. (line 20, page 11)

**RESPONSE TO REVIEWER #2**

**General Comments**

**The manuscript provides the influence of ambient humidity on benzene measurements carried out with an automated in situ GC-PID that is the instrument usually used in air quality monitoring networks. I consider that is an interesting study and therefore, I recommend the manuscript for publication in Atmospheric Measurement Techniques. Some specific comments are given below to be clarified by authors before publication.**

Thank you for this and your valuable suggestions.

**Specific Comments**

**Pag. 2 27. What is the brand, model of the dynamic dilution system? Please include in the text.**

Our dynamic dilution system was built in-house. This has been clarified in the text (line 4, page 3). All the description in the text regarding the humidifier, the flow rate and temperature control refers to this in-house built system. Some of its parts (such as the humidifier) were designed by the authors and some of them were bought (such as the mass flow controllers).

**Pag. 3, line 18. Is the mixture benzene in nitrogen a SRM, if so please indicate it. Who is the supplier of the gas cylinder? Which is the purity? Please, indicate it in the text.**

The mixture of benzene in nitrogen was provided by Abelló Linde. Its concentration was certified by Linde according to the Standard ISO 6141:2007. The expanded uncertainty of the amount fraction of benzene in the mixture was 5%. This information has been added to the manuscript (line 24 on in page 3).

**Are the flow meters calibrated periodically? How?**

The mass flow controllers are at least annually calibrated against a Gilian Gilibrator (a NIOSH primary standard air flow calibrator) available in our lab. If the deviation of the MFC measurements is higher than 1% then the measurements are corrected with a calibration line. If lower, this bias is accepted and accounted for in the uncertainty assessment of the concentration of the gas mixtures.

**Pag. 4, line 10. Authors refer to Standard EN 14662-3:2005 in order to calculate the coefficient brh to see the influence of relative humidity but this standard is already cancelled and the equation 2 does not appear in the EN 14662-3:2015. Why do authors think that this equation has been removed from the updated Standard?**

We think that tests were likely changed because of the non-conformities of analysers with the Standard criterion for the span concentration. It is true that the span concentration (40 µg/m³) is quite high and such high levels are not usually found in the ambient air if measured away from the emission sources. However, from our point of view, it is interesting to test other concentrations apart from the limit value. From our perspective, testing this influence

with a value in between the annual limit and the span (around 20 µg/m$^3$) would be interesting to detect important biases at high benzene amount fractions, which can be present, for instance, in industrial areas. These has been stated in lines 17 on in page 6.

**If the relative humidity is so important at high concentration of benzene as demonstrated in the manuscript, why it has been deleted?**

I am afraid we cannot answer this question. We guess that manufacturers may have been having issues with passing this test and, arguing that ambient concentrations of benzene are not usually as high as 40 µg/m$^3$, they may have put pressure on normalizers to soften the performance criterion.

**Table 1. if (0.01) is the standard deviation, please indicate it in the table.**

It is the standard deviation. This information has been added in Tables 1 and 2.

**Table 3 . A "("should be before µg/m3) in Cmeas. What is qt? in the text are defined qz and qH2O but not qt.**

$q_t$ is the total flow rate, that is, the sum of $q_z$, $q_{H2O}$ and the flow rate from the reference gas mixture. This has been specified in the caption of Table 3.

**On the other hand, have the authors checked the memory effect when using high concentration of benzene (40 ug/m3)?**

Indeed, we did. We followed standard EN 14662-3:2005 for this test (standard EN 14662-3:2016 does not substantially change this test). We measured a nominal concentration of benzene in air of 45 µg/m$^3$ and then we switched to zero air. We carried out this test for Analyser I and II. For Analyser I we obtained the following 3 sequential measurement results: 46.59 µg/m$^3$, -0.01 µg/m$^3$ and -0.01 µg/m$^3$. For Analyser II we obtained: 45.05 µg/m$^3$, 0.32 µg/m$^3$ and 0.31 µg/m$^3$. Both analysers met the performance criteria of the Standard for this test (a measurement < 20% of the annual limit value for analysis n.2 (the one right after changing the concentration to zero) and <10% of the annual limit value for analysis n.3). Analyser II measured 0.05 µg/m$^3$ in a fourth analysis and 0.01 µg/m$^3$ in a fifth one, so we can conclude that this effect is not significant in this type of instruments. In any case, every time we changed the concentration of benzene we excluded the first measurement from the data analysis.

**Technical comments**

**Pag. 1, line 25: revise "h0ematologic"**

Thanks for spotting this. (line 1, page 2)

**Pag. 1 , line 29. Reference properly the Standard EN in reference section. The standard number is missing.**

We have added the whole number of the standard in the text (line 5, page 2).

**Pag. 4. , line 7 and pag. 5, line 4. I think that authors should change "2005" by "2015".**

We have explained that the tests were carried out according to Standard EN 14662-3:2005 as these were carried out before the publication of the new standard. We have also described the differences between the two standards regarding the humidity influence. (line 22 in page 4, and lines 7 on in page 5).

**Pag. 4, line 18. Section 2.2.2.1 should be 2.2.2.2**

Thanks for spotting this.

**Pag. 5, line 6. Revise the sentence: Moreover, coefficient. . .. In air.**

This sentence has been reworded to "Moreover, calculated coefficients $b_{rh}$ for both analysers turned out to be higher than 4% (the maximum variation allowed in the Standard EN 14662-3:2005) when the tests were carried out with 40 $\mu g/m^3$ of benzene in air". This has been accordingly changed in the manuscript (line 11, page 6).

[revised manuscript text omitted]

**Response to Reviewer#1**

First of all, we would like to thank reviewer#1 for taking the time to review our paper again and for his/her valuable suggestions, which have definitively improved our paper.

**Review of the manuscript entitled "Influence of sample temperature and environmental humidity on measurements of benzene in ambient air by transportable GC-PID", by Romero-Trigueros et al.**

**General comments.**

**The manuscript by Romero-Trigueros et al. adress an important issue regarding air quality monitoring of benzene by transportable GC-PID. The tests are reported clearly and the implications discussed appropriately. My main concerns have been addressed in an appropriate way. I recommend the manuscript in its present form for publication in AMT, even if I spotted a few instances where the manuscript could still be improved slighltly. See my comments below.**

**Specific comments.**

- **page 2, lines 6-7: I suggest to use "selective" rather than "specific".**
This change has been addressed.

- **page 3, line 23 (equation (1)): What are the numbers 28.8 and 18 and what are their units?**
These are the molar masses of dry air and water in g/mol, respectively. This explanation has been added to the text. When changing units from mol to litre (for instance, by multiplying and dividing by 22.4 litres in normal conditions, the units of both sides of the equation agree).

- **page 6, section 3.2: Because b_rh is defined twice, I suggest for clarity to rename the latter one b_rh(new) or at least differentiate it in some way (also the units are different).**
Thanks for this suggestion, which has been implemented.

- **page 6, lines 17-19: Is there any reason why the suggested test has not been performed?**
This sentence refers to the new version of Standard EN 14662. In the old version two different concentrations were tested (0.5 and 40 µg/m$^3$) but in the new one, tests are performed only at 5 µg/m$^3$. As it is discussed later in the paper, the higher the benzene concentration the higher the interference for a given water vapour amount fraction. That is why we suggest performing these tests at higher benzene amount fraction. One can argue that 40 µg/m$^3$ is too high and that is why we suggest an intermediate concentration around 20 µg/m$^3$. In order to clarify this discussion in the paper, we have added a sentence in page 5 line 16 to catch the attention of the reader in the fact that the higher concentration of benzene has been removed from the tests. Later, in page 6 we have reworded the sentence starting in line 17 to make our point clearer.

- **page 15, Table 1: The given benzene concentrations in the first column are calculated from the standard concentration and dilution (nominal). What is the uncertainty on these values?**
The expanded uncertainties of the reference values have been added to Table 1.

- **page 16, Table 2: The authors could consider introducing the newer b_rh values as well**.
The value of the coefficient according to EN 14662 has been added in Table 2.

**Technical corrections**

**- page 2, lines 6-7: Fix the use of hyphens/dashes**.

**- page 2, line 7: Fix the reference ("Liaud et al. (2014) recently...").**
**- page 5, line 5: No comma after "(AH)" and replace "from 0 and 32" by "from 0 to 32".**
**- page 6, line 16: Missing full stop after first parenthesis.**
**- page 9, line 15, Fix the use of hyphens/dashes.**
**- page 9, lines 18-19: Avoid paragraphs with only one sentence.**

Thanks for spotting all these typos.

**- page 10, lines 20/30: Use a consistent abbreviation for "AIHAJ" (page 12, lines 7-8).**
In order to be consistent also with the rest of the references, we have opted for *Am. Ind. Hyg. Assoc. J.*

**- page 11, line 11: Is there also a link available? (state the date of last access).**
EN Standards are not public, they have to be bought. In our case, our institution has access to read them online but this is not possible for everybody. For this reason, we think it is better not to include a link.

**- page 11, line 13: State the date of last access.**
This date has been included.

**- page 12, line 20: Fix the name "Mar??nez".**
This has been fixed and changed to Martínez.

**- page 14, Figure 2: I suggest to use E and AH for the equations, as in the main text, and to use the same amount of significant digits.**
The x- and y-axis titles have been changed to AH and E as suggested.

**- page 18, Table 3: Include the name of the variables in the table caption.**
The name of the variables not stated in the original caption have been included as well.

[revised manuscript text omitted]

| | | | | | | | |
|---|---|---|---|---|---|---|---|
| 1.8 | 0.706 | $5.88 \cdot 10^{-3}$ | $8.56 \cdot 10^{-3}$ | 9.466 | 40.05 | 40.09 | 0.10 |
| 25 | 9.604 | 0.0080 | 0.117 | 9.574 | 39.60 | 33.83 | -14.56 |
| 38 | 14.90 | 0.0125 | 0.182 | 9.640 | 39.33 | 27.82 | -29.28 |
| 49 | 19.21 | 0.0161 | 0.235 | 9.693 | 39.12 | 26.95 | -31.10 |
| 70 | 27.24 | 0.0229 | 0.333 | 9.791 | 38.72 | 22.38 | -42.20 |
| 78 | 30.57 | 0.0257 | 0.374 | 9.832 | 38.56 | 20.28 | -47.41 |

**Table 3: Relative (RH) and absolute (AH) humidity, mass fraction of water vapour in the dry air (W), reference concentration of benzene (C$_{ref}$), average reading of analyser I (C$_{meas}$) and relative difference between measurements and reference values (E). q$_t$ is the total flow rate, calculated as the sum of the flowrate of zero air (q$_z$), water vapour (q$_{H2O}$) and the flow rate from the benzene reference gas mixture (q$_{std}$). C$_{std}$ is the amount fraction of benzene in the gas cylinder.**